# Knowledge and awareness of human papillomavirus infection and human papillomavirus vaccine among Kazakhstani women attending gynecological clinics

Torgyn Issa[1◉], Aisha Babi[1◉], Alpamys Issanov[2◉], Ainur Akilzhanova[3], Kadisha Nurgaliyeva[4], Zauresh Abugalieva[4], Azliyati Azizan[1,5], Saleem A. Khan[6], Chee Kai Chan[1,7], Raushan Alibekova[2], Gulzhanat Aimagambetova[1] *

1 Department of Biomedical Sciences, School of Medicine, Nazarbayev University, Nur-Sultan, Kazakhstan,
2 Department of Medicine, School of Medicine, Nazarbayev University, Nur-Sultan, Kazakhstan,
3 Laboratory of Genomic and Personalized Medicine, Center for Life Sciences, National Laboratory Astana, Nazarbayev University, Nur-Sultan, Kazakhstan, 4 Republican Diagnostic Center, University Medical Center, Nur-Sultan, Kazakhstan, 5 College of Osteopathic Medicine, Touro University Nevada, Henderson, Nevada, United States of America, 6 Department of Microbiology and Molecular Genetics, School of Medicine, University of Pittsburgh, Pittsburgh, Pennsylvania, United States of America, 7 College of Science and Technology, Wenzhou-Kean University, Wenzhou, China

◉ These authors contributed equally to this work.
* gulzhanat.aimagambetova@nu.edu.kz

**Data Availability Statement:** The study questionnaires and raw data are available via the

## Abstract

Cervical cancer remains one of the top causes of cancer-related morbidity and mortality all over the world. Currently, however, there are no published studies to assess the knowledge of HPV and cervical cancer in Kazakhstan. This study aimed to assess the awareness of HPV, the knowledge of HPV as a cause of cervical cancer, and the awareness of HPV vaccination among Kazakhstani women visiting gynecological clinics across the country. In addition, the study aimed to identify the factors associated with the awareness of HPV and the HPV vaccine and knowledge of HPV as a major cause of cervical cancer. This was a cross-sectional survey-based study with 2,272 women aged between 18–70 years attending gynecological clinics, who were administered paper-based questionnaires. Data analysis included descriptive statistics consisting of mean values, standard deviations, and frequencies, where applicable. Differences in categorical variables between groups were analyzed using the Chi-square test with a significance value of <0.005. Crude odds ratio (OR) and adjusted odds ratio (AOR) with 95% corresponding confidence intervals were calculated in regression analysis using univariate and multivariable logistic regression models. The mean age of participants was 36.33±10.09 years. More than half (53%) of the participants had been screened for cervical cancer. Among those who were aware of HPV, 46% knew that HPV causes cervical cancer and 52% were aware of the HPV vaccine. The key factors related to outcome variables were age, ethnicity, education, family, number of deliveries, and menarche. From a subgroup analysis, results from the HPV test and Pap smear test were factors related to dependent variables such as awareness of HPV and awareness of HPV vaccination.

link: https://zenodo.org/record/4600664#.
YHBChs9xfIV.

**Funding:** This study was supported by the Faculty
Development Research Grant Program 2019-2021
(Funder Project Reference: 110119FD4528, title: A
molecular epidemiological study to determine the
prevalence of oncogenic HPV strains for CC
prevention in Kazakhstan). The funder had no role
in study design, data collection and analysis,
decision to publish, or preparation of the
manuscript. GA. is a PI of the project. The funder
provided support in the form of salaries for authors
T.I. and A.B., but did not have any additional role in
the study design, data collection and analysis,
decision to publish, or preparation of the
manuscript. The statements relating to the author
contributions were reviewed and the authors'
contributions were accurately indicated. The
specific roles of these authors are articulated in the
'author contributions' section.

**Competing interests:** The authors have declared
that no competing interests exist.

## Introduction

The Human papillomavirus (HPV) causes one of the most common sexually transmitted
infections in the world [1]. There is an estimate that 80% of people will acquire HPV in their
lifetime [2]. The prevalence of HPV was reported to vary from 7 to 14% in the general popula-
tion [3]. Out of numerous types, HPV-16 and HPV-18 are causally associated with 70% of cer-
vical cancer cases and precancerous cervical lesions [1].

Cervical cancer is the fourth most common cancer among women worldwide [4]. In 2018,
there were 570,000 cases of cervical cancer and around 311,000 deaths from cervical cancer
[5]. Moreover, according to the World Health Organization (WHO), by 2030 it is expected
that the number of new cervical cancer cases will reach 700,000. At the same time, the absolute
number of deaths from cervical cancer will reach 400,000 [1]. Therefore, based on such a sig-
nificant annual increase in the number of cases and deaths, cervical cancer represents a major
global public health challenge.

In 2020, WHO launched a global program to eliminate cervical cancer that was announced
earlier, in 2018 [6]. The major steps highlighted in WHO's Global Strategy to Accelerate the
Elimination of Cervical Cancer were vaccination, screening, and treatment [7]. To reduce
cases of cervical cancer, WHO has assigned all countries in the world to meet the require-
ments: (1) 90% of young females should be vaccinated against HPV before the age of 15; (2)
70% of women should be screened for cervical cancer using high-performance tests by the age
of 35, and repeatedly by the age of 45; (3) 90% of women who are diagnosed with cervical neo-
plastic lesions should receive appropriate treatment [1, 6].

Kazakhstan, which is a Central Asian country with a land area of more than 2,724 thousand
km$^2$ and is the ninth-largest country in the world, recently has been classified by the World
Bank as an upper-middle-income country [8]. The population of the country is estimated at
almost 19 million. After obtaining independence in 1991, Kazakhstan has undergone a drastic
transformation in the healthcare system [9]. In Kazakhstan as of today, there are no published
studies on the prevalence of HPV at the country level. However, some data on HPV prevalence
is available from regional and several pilot studies on this topic [10, 11]. However, the results
of these studies cannot be generalized for the whole country, as it has a large territory with a
diverse population.

There is also a lack of local research to determine the statistics of cervical cancer. According
to the HPV Information Centre report, cervical cancer ranks as the second leading cause of
cancer among women and cancer-related death in Kazakhstan. Cervical cancer remains the
second leading cause of female cancer and cancer-related deaths in Kazakhstani women with
over 1,700 new cervical cancer cases diagnosed annually [12, 13]. In comparison with global
estimates, cervical cancer statistics remain high in Kazakhstan.

Cervical cytology remains the main screening method for cervical cancer in Kazakhstan. From
2017 until the current days, the screening is performed by Papanicolaou test (Pap test), and the
screening covers all women between the age of 30–70 years [14]. Although plans for 2019–2020
included HPV genotyping as a part of the cervical cancer screening program [14], HPV testing is
only available on a commercial basis. Despite the free nature of the cervical cancer screening pro-
gram in the country, screening coverage remains low. Between 2008 and 2016, 4,460,320 women
from the whole country had undergone screening for cervical cancer [15]. During these nine
years, there was a 32% decrease in the number of women who underwent screening [15].

HPV vaccines (Gardasil and Cervarix) were introduced and administered through a pilot
program in four regions of Kazakhstan in 2013. However, the absence of an HPV vaccination
communication plan, the media's negative coverage of these cases, combined with insufficient
training of health care workers, had a negative impact on the public's willingness to receive the

vaccine. The vaccination coverage was suspended in 2015 even when, at the time, there are more than 6 million women still at risk of cervical cancer in the country [12, 16].

The success and benefit of control and prevention of cervical cancer largely depend on the level of awareness and knowledge about different aspects of the disease and the vaccine. Such low coverage of free cancer screening program and the lack of free HPV testing could be related to the lack of knowledge in the general population regarding HPV and cervical cancer. It is important to mention that WHO's Global Strategy to Accelerate the Elimination of Cervical Cancer not only addresses the importance of cervical cancer screening but also the importance of HPV vaccination as a prevention strategy for this disease. Moreover, with the plan for the HPV vaccine introduction on the national level in Kazakhstan [14], the education and awareness among the general population became even more significant for the success of the program. Knowledge and understanding of behavioral perception towards HPV vaccination are crucial for the development of an effective health policy that will support HPV vaccination among the population [17].

This study aims to assess the awareness of HPV, the knowledge of HPV as a cause of cervical cancer, and the awareness of HPV vaccination among Kazakhstani women visiting gynecological clinics across the country. Moreover, the purpose of this study is to identify the factors, such as social and demographic characteristics, and gynecological health characteristics of the participants, associated with the awareness of HPV and HPV vaccine, and knowledge of HPV as a major cause of cervical cancer.

## Materials and methods

### Study participants and sample collection

A cross-sectional study among women from five cities of central (Nur-Sultan, the capital city), southern (Almaty), western (Aktobe), northern (Pavlodar), and eastern (Oskemen) parts of Kazakhstan was conducted from May 25, 2019, until December 2020. Women aged between 18 and 70 attending gynecological clinics were recruited to the study by the convenience sampling method.

### Study instrument

There were two types of questionnaires utilized to collect data on patients' medical gynecological history and awareness and knowledge of HPV, and awareness of HPV vaccination (S1 File). The first questionnaire (S2 File) was filled out by doctors and consisted of 30 items and included the following patient information: socio-demographic characteristics, lifestyle characteristics, and the history of gynecological diseases.

The second questionnaire (S3 File) that was filled out by the participants was adapted from previous studies [18, 19]. The patient questionnaire consisted of 25 items and included the following information about the patients: socio-demographic characteristics, awareness of cervical cancer and the associated risk factors, awareness of screening for cervical cancer, whether the patient had gone through screening for cervical cancer, awareness of HPV, awareness of risks of HPV, and awareness of HPV vaccine. Only those who were aware of HPV continued to answer the questions about knowledge of HPV and awareness of the HPV vaccine. The survey was conducted in Kazakh and Russian languages depending on the preferences of the participants.

### Sample collection and genotyping

All the study participants had Papanicolaou (Pap smear) testing done. The cervical swabs were used as a source of clinical samples for HPV genotyping, which was performed in the laboratory using the AmpliSens kit; this test is designed to identify 14 high-risk HPV types which are

16, 18, 31, 33, 35, 39, 45, 51, 52, 56, 58, 59, 66, 68. The swab samples were collected into 1.5 mL Eppendorf tubes by gynecologists using cytobrush and were transported and stored in a frozen state (-20°C) until needed for the DNA extraction procedure. DNA was extracted using the Wizard® Genomic DNA Purification Kit according to the manufacturer's protocol. Real-time PCR analysis was performed using the CFX 96 Real-Time PCR detection system from the Bio-Rad Laboratories. Each PCR run had positive, negative, and internal controls, as per the manufacturer's instructions. Genotyping results were divided into two outcomes; the first is positive, which isa sample having any of the 14 high-risk HPV types and the second outcome is negative meaning that the sample does not have any of these HPV types. All HPV positive women were invited for follow-up visits and treated according to the national guideline for cervical lesions management.

## Study variables

Independent variables were socioeconomic and demographic characteristics (age, ethnicity (Kazakh, Russian and other ethnicities), and city of residence) of the participants. Also, information on marital status (not single and married, in a committed relationship; single, either single, widowed or divorced), and family (number of children, history of delivery, history of abortion) characteristics of participants were collected. Gynecological health (age at the start of sex life, menarche, result of cervical cytology test by Pap smear, result of HPV genotyping test) were collected.

Outcome variables for this study were the following: awareness of HPV ("Have you ever heard about the HPV vaccine?" Yes/No); knowledge of HPV as a major cause of cervical cancer ("HPV infection is the major cause of cervical cancer." True/False); and awareness of HPV vaccine ("There is a vaccine against HPV infection." True/False).

## Ethical considerations

The ethical approval to conduct this study was given by the Institutional Research Ethics Committee of Nazarbayev University (NU-IREC) on April 23, 2019 (IREC decision number: 146/ 4042019). Study participants were informed about the risks, benefits, goals, and methods of the study. After receiving verbal consent, study participants responded to the survey questions.

Particularly for this research, verbal consent was used as a way to receive informed consent from a participant. As anonymity of the study participants was the primary concern, verbal consent was deemed as most appropriate. Moreover, no personal information related to any of the patients was made available to the Investigators at any time before, during, or after the study. All the information about the study and participants' rights was stated both orally and on the information letter provided to the participants.

It is important to mention the cultural context. Verbal consent was prioritized as Kazakhstan is a post-Soviet country where people still are not comfortable with signing documents that are similar to a contract. Moreover, the general population tends to have low trust towards interviews and researchers. Such a low trust in the confidentiality of research is most common among the older generation, and our study recruited participants until the age of 70.

Verbal consent used for this study included all the necessary components for informed consent. First of all the doctor explained the title, research itself, and the major goals of the research. Secondly, the informed consent included detailed information on how long the participation will take time and what kind of questions are included and asked. Thirdly the verbal consent included the information on patient confidentiality, patient risks and rewards. Lastly the informed consent included the contacts of the Principal Investigator of the research; in case patients had any additional questions to ask in the future.

## Data analysis

Statistical analysis was performed using STATA 16 [20]. Data analysis included descriptive statistics consisting of mean values, standard deviations, and frequencies, where applicable. Relationships between categorical variables were analyzed using the Chi-square test with a significance value of <0.005. Crude odds ratio (COR) and adjusted odds ratio (AOR) with 95% corresponding confidence intervals were calculated in univariate and multivariable logistic regression models. A significance value <0.05 was used as an indication of an association between variables.

## Results

### Participant characteristics

The total number of participants recruited for the study was 2,272 with approximately an equal number of participants each of the five cities involved. The descriptive characteristics of the study participants are shown in Table 1. Study participants were diverse in terms of age, ethnicity, educational level, city, family, age of the first menarche, age during the first intercourse, number of living children, and number of deliveries, number of abortions, Pap test result, and HPV status. The mean age of the study participants was 36.33±10.09 with a range of 18–70 years old. More than three-quarters (75%) of women belonged to the Kazakh ethnicity. Almost half of women (48%) had undergraduate and/or graduate university degrees. The majority of the participants were in a committed relationship (81%). More than three-quarters of women have had one or more children. The majority of women (92%) had normal Pap test results and more than half of the participants had negative results of the HPV genotyping test.

### HPV awareness

As summarized in Table 2, 53% of the respondents were informed about HPV. Bivariate analysis using the Chi-square test showed that HPV awareness was significantly associated with age, ethnicity, education level, city, family groups, menarche groups, and the number of deliveries, Pap test results, and HPV status.

In the multivariable logistic model, the following factors which were positively associated with HPV awareness was higher in age group of 36–45, having a university degree, and being in a committed relationship.

Women aged between 36–45 who had one or more deliveries, were 1.55-times more likely to be aware of HPV in comparison with women aged 18–25 who had one or more deliveries (Table 2). The subgroup analysis among women who had Pap test and HPV test showed different results in the awareness about HPV (Table 2). The following factors were positively associated with HPV awareness; age group of 36–45, being of russian ethnicity, having a university degree, and having positive results of HPV test.

The sources from which the study participants heard about HPV are shown in Fig 1. The majority of women who were aware of HPV received this information from their gynecologist. The second most common source was the Internet (17%). The third and fourth most common sources of HPV information were general practitioners (7%) and television (7%). The rest of the sources (5%) included other health professionals, friends and peers, family members, educational settings, magazines/books, and nurse practitioners.

### HPV knowledge

Among those who were aware of HPV (N = 1,124), less than half of the respondents (46%) knew that HPV is the major cause of cervical cancer (Table 3). Bivariate analysis using the

**Table 1. Demographic and clinical characteristics of women visiting gynecological clinics, (N = 2,272).**

| Variables | Total N = 2,272, (%) |
|---|---|
| **Age,** mean 36.33±10.09 | |
| **18–25** | 310 (14%) |
| **26–35** | 896 (39%) |
| **36–45** | 635 (28%) |
| **46+** | 431 (19%) |
| **Ethnicity** | |
| **Kazakh** | 1713 (75%) |
| **Russian** | 420 (19%) |
| **Other** | 139 (6%) |
| **Education** | |
| **Unfinished/finished school** | 399 (17%) |
| **College** | 788 (35%) |
| **University** | 1085 (48%) |
| **City** | |
| **Nur-Sultan** | 352 (15%) |
| **Almaty** | 495 (22%) |
| **Aktobe** | 490 (22%) |
| **Oskemen** | 462 (20%) |
| **Pavlodar** | 473 (21%) |
| **Family** | |
| **Single** | 423 (19%) |
| **Not single** | 1849 (81%) |
| **Age of menarche,** mean 13.60±1.41 | |
| **<13** | 348 (15%) |
| **13–15** | 1386 (61%) |
| **>15** | 538 (24%) |
| **Age of first intercourse,** mean 20.47±2.97 | |
| **< 18** | 221 (10%) |
| **= >18** | 2051 (90%) |
| **Number of alive children** | |
| **0** | 455 (20%) |
| **≥1** | 1817 (80%) |
| **Number of deliveries** | |
| **0** | 439 (19%) |
| **≥1** | 1833 (81%) |
| **Number of abortions** | |
| **0** | 1095 (48%) |
| **≥1** | 1177 (52%) |
| **Pap test status (N = 1841)** | |
| **Normal** | 1702 (92%) |
| **Abnormal** | 139 (8%) |
| **HPV status (N = 759)** | |
| **Negative** | 466 (61%) |
| **Positive** | 293 (39%) |

**Table 2. Awareness of HPV among women visiting gynecological clinics, (N = 2,272).**

| Variables | Aware of HPV N = 1,213; 53.39%, %, p-value | OR (95% CI) N = 2,272 | AOR (95% CI)* N = 2,272 | | AOR subgroup (95% CI)** N = 759 |
|---|---|---|---|---|---|
| **Age** | | | No deliveries | One or more deliveries | |
| **18–25** | 160 (52%) | 1 | 1 | 1 | 1 |
| **26–35** | 455 (51%) | 0.96 (0.75–1.25) | 1.18 (0.76–1.84) | 0.96 (0.67–1.37) | 1.01 (0.60–1.71) |
| **36–45** | 369 (58%) | 1.30 (0.99–1.71) | 0.73 (0.39–1.34) | 1.55 (1.07–2.23) | 1.37 (0.78–2.42) |
| **≥46** | 229 (53%) | 1.06 (0.79–1.42) | 1.44 (0.62–3.33) | 1.28 (0.87–1.87) | 1.03 (0.55–1.92) |
| | p-value = 0.036 | | | | |
| **Ethnicity** | | | | | |
| **Kazakh** | 840 (49%) | 1 | 1 | | 1 |
| **Russian** | 277 (66%) | 2.01 (1.61–2.52) | 1.98 (1.58–2.49) | | 2.11 (1.45–3.08) |
| **Other** | 96 (69%) | 2.32 (1.60–3.37) | 2.21 (1.52–3.23) | | 2.07 (1.11–3.87) |
| | p-value = 0.000 | | | | |
| **Education** | | | | | |
| **Unfinished/finished school** | 181 (45%) | 1 | 1 | | 1 |
| **College** | 390 (50%) | 1.18 (0.93–1.50) | 1.24 (0.97–1.60) | | 0.90(0.56–1.45) |
| **University** | 642 (59%) | 1.75 (1.39–2.20) | 1.82 (1.43–2.31) | | 1.28 (0.83–1.97) |
| | p-value = 0.000 | | | | |
| **Age of menarche,** | | | | | |
| **<13** | 215 (62%) | 1 | 1 | | 1 |
| **13–15** | 758 (55%) | 0.75 (0.59–0.95) | 0.82 (0.64–1.05) | | 0.91 (0.57–1.47) |
| **>15** | 240 (45%) | 0.50 (0.38–0.66) | 0.55 (0.42–0.73) | | 0.73 (0.42–1.26) |
| | p-value = 0.000 | | | | |
| **Number of deliveries** | | | | | |
| **0** | 255 (58%) | 1 | 1 | | 1 |
| **≥1** | 958 (52%) | 0.79 (0.64–0.98) | 0.78 (0.49–1.24) | | 0.82 (0.54–1.25) |
| | p-value = 0.028 | | | | |
| **Pap test status** | N = 1015 | N = 1841 | | | |
| **Normal** | 960 (56%) | 1 | | | 1 |
| **Abnormal** | 55 (40%) | 0.51 (0.36–0.72) | | | 0.47 (0.26–0.88) |
| | p-value = 0.000 | | | | |
| **HPV status** | N = 362 | N = 759 | | | |
| **Negative** | 200 (43%) | 1 | | | 1 |
| **Positive** | 162 (55%) | 1.64 (1.22–2.21) | | | 1.63 (1.19–2.24) |
| | p-value = 0.001 | | | | |

*Adjusted for age, ethnicity, education, age of menarche, number of deliveries

**Adjusted for age, ethnicity, education, age of menarche, number of deliveries, Pap test status, HPV status

Chi-square test showed that there is a statistically significant difference in the knowledge of HPV as a major cause of cervical cancer among ethnicity groups, education levels, and cities. In the multivariable logistic model, the following factors were positively associated with knowledge of HPV as the major cause of cervical cancer: age group of 26–35 being of other ethnic groups, having a university degree and having one or more deliveries.

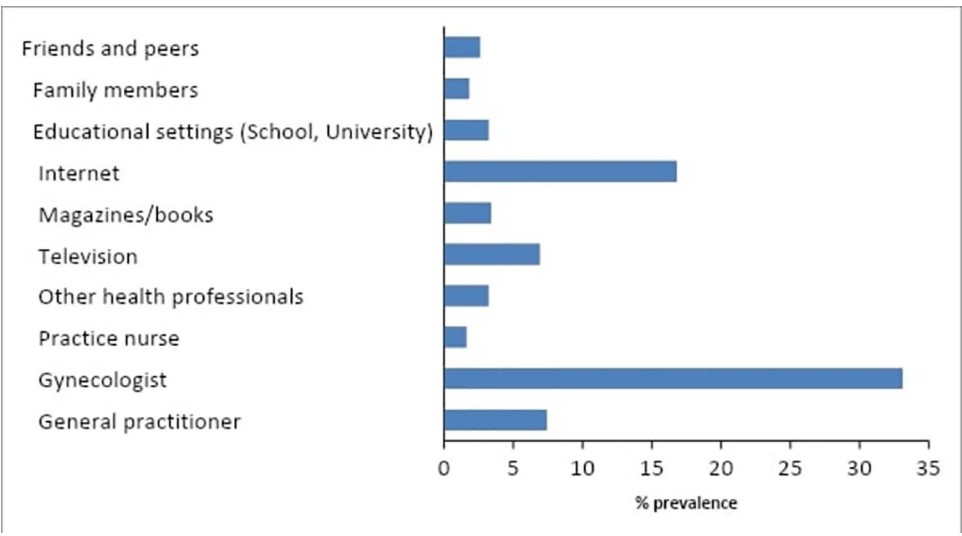

**Fig 1. Sources of information on HPV.**

## HPV vaccination

Among the study participants who were aware of HPV (N = 1,205), half of the respondents (52%) were also aware of the HPV vaccine (Table 4). Bivariate analysis using the Chi-square test showed that there is a statistically significant difference in HPV vaccine awareness within

**Table 3. Knowledge of HPV as the major cause of cervical cancer among women visiting gynecological clinics, (N = 1,124).**

| Variables | HPV is the major cause of cervical cancer | | |
|---|---|---|---|
| | N = 522; 46.44%, %, p-value | OR (95% CI) N = 1,124 | AOR (95% CI)* N = 1,124 |
| **Age** | | | |
| **18–25** | 60 (40%) | 1 | 1 |
| **26–35** | 206 (49%) | 1.37 (0.94–2.00) | 1.27 (0.86–1.89) |
| **36–45** | 160 (47%) | 1.26 (0.85–1.86) | 1.21 (0.80–1.85) |
| **46+** | 96 (46%) | 1.23 (0.80–1.89) | 1.23 (0.79–1.95) |
| | p-value = 0.444 | | |
| **Ethnicity** | | | |
| **Kazakh** | 379 (48%) | 1 | 1 |
| **Russian** | 98 (40%) | 0.71 (0.53–0.94) | 0.74 (0.55–0.99) |
| **Other** | 45 (51%) | 1.13 (0.73–1.75) | 1.13 (0.73–1.77) |
| | p-value = 0.040 | | |
| **Education** | | | |
| **Unfinished/finished school** | 62 (36%) | 1 | 1 |
| **College** | 162 (45%) | 1.47 (1.01–2.14) | 1.41 (0.97–2.06) |
| **University** | 298 (50%) | 1.81 (1.28–2.58) | 1.75 (1.22–2.50) |
| | p-value = 0.003 | | |
| **Number of deliveries** | | | |
| **0** | 104 (44%) | 1 | 1 |
| **≥1** | 418 (47%) | 1.14 (0.85–1.52) | 1.14 (0.83–1.56) |
| | p-value = 0.374 | | |

*Adjusted for age, ethnicity, education, number of deliveries

**Table 4. Awareness of HPV vaccination among women visiting gynecological clinics, (N = 1,205).**

| Variables | Aware of vaccine N = 621; 52.4% %, p value | OR (95% CI), N = 1,205 | AOR (95% CI)*, N = 1,205 | AOR subgroup (95% CI)** N = 1,009 |
|---|---|---|---|---|
| **Age,** | | | | |
| **18–25** | 68 (43%) | 1 | 1 | 1 |
| **26–35** | 219 (49%) | 1.29 (0.90–1.85) | 1.17 (0.81–1.72) | 1.11 (0.73–1.67) |
| **36–45** | 203 (55%) | 1.66 (1.14–2.42) | 1.60 (1.09–2.36) | 1.48 (0.97–2.26) |
| **≥46** | 131 (52%) | 1.83 (1.21–2.75) | 1.90 (1.24–2.88) | 1.78 (1.11–2.83) |
| | p-value = 0.008 | | | |
| **Ethnicity** | | | | |
| **Kazakh** | 411 (49%) | 1 | 1 | 1 |
| **Russian** | 157 (57%) | 1.37 (1.04–1.81) | 1.37 (1.04–1.82) | 1.36 (1.00–1.84) |
| **Other** | 53 (56%) | 1.30 (0.85–2.00) | 1.30 (0.84–2.01) | 1.23 (0.77–1.95) |
| | p-value = 0.053 | | | |
| **Education** | | | | |
| **Unfinished/ finished school** | 85 (48%) | 1 | 1 | 1 |
| **College** | 187 (48%) | 1.03 (0.72–1.47) | 1.07 (0.75–1.54) | 1.23 (0.82–1.83) |
| **University** | 349 (55%) | 1.34 (0.96–1.86) | 1.46 (1.04–2.06) | 1.46 (1.00–2.14) |
| | p-value = 0.065 | | | |
| **Family** | | | | |
| **Single** | 112 (45%) | 1 | 1 | 1 |
| **Not single** | 509 (53%) | 1.39 (1.05–1.84) | 1.40 (1.04–1.86) | 1.30 (0.96–1.78) |
| | p-value = 0.020 | | | |
| **Age of menarche** | | | | |
| **<13** | 121 (57%) | 1 | 1 | 1 |
| **13–15** | 398 (53%) | 0.85 (0.62–1.15) | 0.86 (0.63–1.18) | 0.87 (0.63–1.22) |
| **>15** | 102(43%) | 0.57 (0.40–0.83) | 0.56 (0.38–0.81) | 0.52 (0.34–0.78) |
| | p-value = 0.008 | | | |
| **Pap test status, N = 521** | | | | |
| **Normal** | 499 (52%) | 1 | | 1 |
| **Abnormal** | 22 (40%) | 0.61 (0.35–1.06) | | 0.57 (0.32–1.01) |
| | p-value = 0.076 | | | |

*Adjusted for age, ethnicity, education, family, age of menarche

**Adjusted for age, ethnicity, education, family, age of menarche, Pap test status

age groups, city groups, family groups, and menarche groups. In the multivariable logistic model, the following three factors were positively associated with HPV vaccine awareness: age group of 36–45, as well as 46 and older, being of Russian ethnic group, having a university degree, and being in a committed relationship.

In the multivariable logistic model of a subgroup (N = 1,009), the following three factors were positively associated with HPV vaccine awareness; age group of 46 and older, being of Russian ethnic group, having a university degree(s), and being married or being in a relationship.

## Discussion

This is the first study in Kazakhstan with the aim to assess the knowledge and awareness of HPV infection as a cause of cervical cancer, and the awareness of HPV vaccination among 18-

70-year-old women visiting gynecological clinics in five different regions of the country. In addition, this study aimed to identify the factors associated with the awareness of HPV infection and HPV vaccine, and the knowledge of HPV infection as a major cause of cervical cancer. As cervical cancer causes over 300,000 deaths per year worldwide and is the second leading cause of death from cancer among women in Kazakhstan [5, 12, 13], it is important to implement prevention strategies to reduce the disease incidence. Therefore, information about HPV infection and HPV vaccination is important topics to study as it can help in the prevention of cervical cancer.

The observed level of HPV awareness in this study is similar to the research conducted in Serbia, where cervical cancer is one of the most frequent cancer types among the population. The results of the study conducted among women visiting Serbian cervical cancer counseling center revealed 61% of HPV awareness [21]. The results of HPV awareness from our study are also similar to the results from the studies conducted in other upper middle-income countries. The study from Brazil showed that 40% of respondents were aware of HPV infection [22], while the study conducted in China showed a 51% of HPV awareness [23].

In our study, the major sources of information about the HPV vaccine were gynecologists and general practitioners, the Internet, and television. Results from other studies on HPV awareness confirm that medical workers and media are the main sources of awareness about HPV and cervical cancer [22, 23]. Such a relatively high prevalence of HPV awareness can be attributed to the fact that healthcare in Kazakhstan is free, and therefore there are fewer barriers for women to visit medical facilities.

Results of our study are similar to the research conducted in China where knowledge of HPV as a cause of cervical cancer is found to be about 40% [23]. It is important to note that awareness of HPV was relatively higher than the knowledge of HPV. Therefore, a high level of awareness of HPV does not necessarily lead to a high level of knowledge about HPV. Relatively low level of vaccine awareness reported in our study is a common trend in other countries as well. In Serbia there was only a 23% of HPV vaccine awareness, in China only 16% of women were aware of the HPV vaccine [21, 23].

One important finding of this study shows that level of education is positively associated with HPV awareness, knowledge of HPV as the major cause of cervical cancer, and awareness of the HPV vaccine. This finding is similar to the results of the study conducted in the United States, which found that adults with college degrees are more likely to be aware and knowledgeable about the HPV infection and the HPV vaccination [24].

This study also found ethnicity as a positive factor for all three outcome variables. Findings regarding ethnic groups demonstrate that Kazakh women are less aware of the HPV infection and the HPV vaccination. This might be explained by the fact that the majority of the sources of information about HPV and HPV vaccination are communicated in Russian. Therefore, women who can only speak Kazakh have limited access to this information.

Having one or more births was found to be a positive factor for HPV knowledge. This can be explained by the fact that women having the deliveries were more likely to visit the hospital and while there, they will get information about HPV infection.

Regarding awareness of the HPV vaccine, being married or being in a committed relationship was found to be a positive factor. Women who were not single were more likely to be aware of the HPV vaccine. This can be attributed to the fact that women who were married or in a relationship were more likely to have a child and therefore would visit the hospital where they would obtain the information about HPV vaccination.

Having menarche after 15 years of age was found to be a negative factor for all three outcome variables. Studies showed that girls having menarche in earlier ages before 12 tend to be interested in sexual life at earlier ages [25]. Therefore, women who had menarche after 15 were

less likely to have sexual relations at an earlier age; as a result, they were less aware of sexually transmitted infections such as HPV.

Subgroup analysis of women that had HPV and Pap smear tests showed that HPV test with a positive result is a positive factor for HPV awareness, and Pap smear with an abnormal result is a negative factor for HPV awareness. Pap smear test with an abnormal result was also a negative factor for the awareness of HPV vaccination. Pap smear test is not a direct test to detect HPV, but rather is a test to check the smear for abnormal cytology of cervical cells; therefore, women who had Pap smear testing were not necessarily aware of HPV and HPV vaccination. However, undergoing an HPV test means that women will at least have heard the word HPV itself thus may be curious to learn more about the virus and why this test was conducted.

This is the first study to evaluate the level of knowledge and awareness of women in Kazakhstan on the topics of HPV and HPV vaccination, which covers majority of the country's regions. The study demonstrates a limited knowledge among the public whereby half of the women were aware of HPV infection, and only a quarter knew of the link between HPV and cervical cancer, and only a quarter of interviewed women were aware of the vaccine against HPV.

## Strength and limitations

Due to the design of this study, no causality can be established between the chosen factors and level of knowledge and awareness of HPV and HPV vaccination. Moreover, the convenient sampling method used in this study introduced selection bias. We did not have socio-demographic characteristics of women who did not participate in our study but also attended gynecological clinics, so we could not determine whether there is a significant difference between respondents and non-respondents. Considering the self-reporting nature of the participants' answers, this could also result in recall bias, underreporting or exaggeration of the responses.

Although our study had more than two thousand participants from the north, south, east, west, and center of the country, the results of our study cannot be applied to all women in Kazakhstan. Sampling through convenience in gynecological offices could exclude certain demographic groups. Moreover, despite a huge migration of the rural population to the cities, rural areas of the Kazakhstan can differ significantly from the cities where the study took place. Therefore, a sizable portion of the female population is not represented in the study.

## Clinical implication

To decrease the incidence of cervical cancer in Kazakhstan all existing technologies such as HPV infection testing, effective cervical cancer screening, colposcopy examination, and vaccination should be employed [1, 6]. However, the success of the governmental programs for cervical cancer prevention will depend on the population's knowledge and awareness of the problem and the ways of prophylactics. In this view, population-based information programs to enhance the knowledge and awareness, thus increasing understanding of the risks related to HPV infection and its association with cervical cancer could enhance the public's acceptance of the screening program. The screening program should be more carefully implemented and reinforced to reach a demand level of 70% [1].

The national HPV vaccination program should be restarted after careful education of the population and proper introduction. Before introducing the HPV vaccine into the national immunization schedule and to increase possible vaccination uptake, it is crucial to conduct research to better understand the population's knowledge, awareness, and understanding barriers to vaccination, attitudes, practices, and beliefs about immunization specifically regarding the HPV vaccine. The task appears even more complex due to the challenges faced during the

previous HPV vaccination effort in Kazakhstan, along with the current public hesitancy towards vaccines in general. The overall goal is to work toward informing the population in terms of best practices that should be adopted with regards to screening and vaccination prevention programs. This would result in a significant reduction in the incidence and prevalence of cervical cancer in Kazakhstan and will help the country to achieve the goals set by the WHO.

## Conclusions

The results of this cross-sectional study suggests that the awareness of HPV among women visiting gynecological clinics in Kazakhstan is relatively high. However, the knowledge about HPV as a cause of cervical cancer and awareness of HPV vaccination should be improved. As the success of the governmental programs for cervical cancer prevention largely depends on the population's knowledge and awareness of HPV and cervical cancer, there is an urgent need for educational intervention; both formal and informal. Population-based informational programs need to enhance the knowledge and awareness, which would increase understanding of the risks related to the HPV infection and its association with cervical cancer. Overall, this could improve the public's acceptance of the cervical cancer screening program in Kazakhstan.

## Supporting information

**S1 File. Survey protocol.**
(DOCX)

**S2 File. Clinical data questionnaire.**
(DOCX)

**S3 File. HPV survey.**
(DOCX)

## Acknowledgments

The author would like to acknowledge the Nazarbayev University School of Medicine for the supportive atmosphere that enabled the completion of this study.

## Author Contributions

**Conceptualization:** Azliyati Azizan, Saleem A. Khan, Chee Kai Chan, Gulzhanat Aimagambetova.

**Data curation:** Torgyn Issa, Aisha Babi, Alpamys Issanov, Ainur Akilzhanova, Kadisha Nurgaliyeva, Zauresh Abugalieva, Chee Kai Chan.

**Formal analysis:** Torgyn Issa, Aisha Babi, Alpamys Issanov, Zauresh Abugalieva, Chee Kai Chan, Raushan Alibekova.

**Funding acquisition:** Gulzhanat Aimagambetova.

**Investigation:** Alpamys Issanov, Ainur Akilzhanova, Kadisha Nurgaliyeva, Zauresh Abugalieva, Azliyati Azizan, Chee Kai Chan, Gulzhanat Aimagambetova.

**Methodology:** Azliyati Azizan, Raushan Alibekova, Gulzhanat Aimagambetova.

**Project administration:** Torgyn Issa, Gulzhanat Aimagambetova.

**Resources:** Gulzhanat Aimagambetova.

**Software:** Alpamys Issanov.

**Supervision:** Ainur Akilzhanova, Kadisha Nurgaliyeva, Zauresh Abugalieva, Saleem A. Khan, Gulzhanat Aimagambetova.

**Validation:** Aisha Babi, Alpamys Issanov, Raushan Alibekova.

**Visualization:** Torgyn Issa, Aisha Babi, Alpamys Issanov.

**Writing – original draft:** Torgyn Issa, Aisha Babi, Alpamys Issanov, Ainur Akilzhanova, Gulzhanat Aimagambetova.

**Writing – review & editing:** Azliyati Azizan, Saleem A. Khan, Chee Kai Chan, Gulzhanat Aimagambetova.

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
