## [Decision Letter · Decision Letter 0]

8 Oct 2021

PONE-D-21-11756Knowledge and awareness of human papillomavirus infection and human papillomavirus vaccine among Kazakhstani women attending gynecological clinicsPLOS ONE

Dear Dr. Aimagambetova,

Thank you for submitting your manuscript to PLOS ONE. After careful consideration, we feel that it has merit but does not fully meet PLOS ONE’s publication criteria as it currently stands. Therefore, we invite you to submit a revised version of the manuscript that addresses the points raised during the review process.

We look forward to receiving your revised manuscript.

Kind regards,

Stanley J. Robboy, MD

Academic Editor

PLOS ONE

Journal Requirements:

4. Please include additional information regarding the survey or questionnaire used in the study and ensure that you have provided sufficient details that others could replicate the analyses. For instance, if you developed a questionnaire as part of this study and it is not under a copyright more restrictive than CC-BY, please include a copy, in both the original language and English, as Supporting Information.

In your Methods section, please provide a justification for the sample size used in your study, including any relevant power calculations (if applicable).

Finally please provide additional information regarding further treatment for women whom where identified as HPV positive during the study (for instane whether they were referred to a medical specialist for further treatment upon knowledge of their diagnosis). 

 "This study was supported by the Faculty Development Research Grant Program 2019-2021 (Funder Project Reference: 110119FD4528, title: A molecular epidemiological study to determine the prevalence of oncogenic HPV strains for CC prevention in Kazakhstan). The funder had no role in study design, data collection and analysis, decision to publish, or preparation of the manuscript. GA. is a PI of the project." 

We note that one or more of the authors is affiliated with the funding organization, indicating the funder may have had some role in the design, data collection, analysis or preparation of your manuscript for publication; in other words, the funder played an indirect role through the participation of the co-authors. If the funding organization did not play a role in the study design, data collection and analysis, decision to publish, or preparation of the manuscript and only provided financial support in the form of authors' salaries and/or research materials, please do the following:

a. Review your statements relating to the author contributions, and ensure you have specifically and accurately indicated the role(s) that these authors had in your study. These amendments should be made in the online form.

b. Confirm in your cover letter that you agree with the following statement, and we will change the online submission form on your behalf: 

“The funder provided support in the form of salaries for authors [insert relevant initials], but did not have any additional role in the study design, data collection and analysis, decision to publish, or preparation of the manuscript. The specific roles of these authors are articulated in the ‘author contributions’ section.

Reviewers' comments:

Reviewer's Responses to Questions

**Comments to the Author**

1. Is the manuscript technically sound, and do the data support the conclusions?

Reviewer #1: Yes

Reviewer #2: Yes

2. Has the statistical analysis been performed appropriately and rigorously? 

Reviewer #1: Yes

Reviewer #2: Yes

3. Have the authors made all data underlying the findings in their manuscript fully available?

Reviewer #1: Yes

Reviewer #2: Yes

4. Is the manuscript presented in an intelligible fashion and written in standard English?

Reviewer #1: Yes

Reviewer #2: Yes

5. Review Comments to the Author

Reviewer #1: This is a very good manuscript. A very interesting and actual topic, which was very clear and well presented by the authors. The study design and statistical analysis are appropriate. The refferences are very well chosed.

However, there are some minor issues that have to be addressed.

1. There are some minor ortographic erors (highlighted in the reviewed manuscript)

2. In the tables is better to present the results as: numbers (and percentage) in order to be more explicit and to have a more real image of the values. For example: 10 (3.25).

3. A cross-sectional study CAN NOT BE a prospective study!!!!!! A cross-sectional study means like "a snapshot".

A prospective study involves following up the subjects over a period of time (longitudinal collection of data).

4. Usually in the "Conclusions" section the authors do not present figures and percentage form the "Results/ Discussions" sections. They present the general conclusions of the study, which were very well highlighted for the current research below the phrase with the figures, in the Conclusions section of the manuscript.

Reviewer #2: Knowledge and awareness of HPV infection and HPV vaccine among Kazakhstani women

Issa et al present the findings from the well done study “Kazakhstani women’s Knowledge and awareness of HPV infection and HPV vaccine.”

The tables and figure are clear and concise.

The text, in contrast, especially in RESULTS is difficult to read and needs to be simplified. For example, is it necessary to repeat the data from the tables, especially that contained in parentheses (e.g., AOR=0.73.; CI: 0.39-1.34, referent=age group of 18-25).

L217: The text states that women age 46 and over were aware of HPV, which I take to mean this is an important finding. To me it then implies that younger women likely are not. Is it then necessary several lines later to actually state that younger women were not.

L235-L240 The text repeats data in results.

The entire RESULTS section should be made more readable

L243: (Among those who were aware about HPV vaccine (N=1,215), less than half of the respondents (46.36%) knew that HPV is the major cause of cervical cancer).

If I interpret the authors’ meaning correctly, fewer than half of the 1215 women who were aware of the vaccine understood what the vaccine treated because they were not aware that HPV is the cause of cervical cancer. If so, the implication the implication for the need for education is important: What is cancer and how does it come about; what are vaccines and what do they treat.

L297-L304: Largely repeats the introduction. It should be truncated or eliminated. Much of the remaining DISCUSSION unnecessarily repeats the results and can be truncated.

The CONCLUSION section repeats too much of the results, but insufficiently says what was learned from the study.

6. PLOS authors have the option to publish the peer review history of their article (what does this mean?). If published, this will include your full peer review and any attached files.

Reviewer #1: **Yes: **MIOARA MATEI

Reviewer #2: **Yes: **Stanley J Robboy

---

## [Author Response · Author response to Decision Letter 0]

4 Nov 2021

Response to the Reviewers 

#1

Dear Reviewer, 

Thank you very much for the careful evaluation of our manuscript. We appreciate your time, efforts, valuable comments and suggestions that helped us to improve the manuscript quality. Please find below our point-by-point responses for all your comments.

Reviewer #1: This is a very good manuscript. A very interesting and actual topic, which was very clear and well presented by the authors. The study design and statistical analysis are appropriate. The refferences are very well chosed.

However, there are some minor issues that have to be addressed.

1. There are some minor ortographic erors (highlighted in the reviewed manuscript)

Response: Thank you very much for the comment. A careful editing of the manuscript have been performed. Errors and typos are removed. Please see the manuscript submitted with the track changes. 

2. In the tables is better to present the results as: numbers (and percentage) in order to be more explicit and to have a more real image of the values. For example: 10 (3.25).

Response: Thank you for the comment. All table were recomposed and the results are presented in numbers (and percentages) as was suggested. Please see the manuscript submitted with the track changes.

3. A cross-sectional study CAN NOT BE a prospective study!!!!!! A cross-sectional study means like "a snapshot".

A prospective study involves following up the subjects over a period of time (longitudinal collection of data).

Response: Thank you or the comment. The error was removed and the sentence now appears as follows: “A prospective cross-sectional study among women from five cities of central (Nur-Sultan, the capital city), southern (Almaty), western (Aktobe), northern (Pavlodar), and eastern (Oskemen) parts of Kazakhstan was conducted from May 25, 2019, until December 2020.”

4. Usually in the "Conclusions" section the authors do not present figures and percentage form the "Results/ Discussions" sections. They present the general conclusions of the study, which were very well highlighted for the current research below the phrase with the figures, in the Conclusions section of the manuscript.

Response: Thank you for the comment. The conclusion section was recomposed as was suggested and now appears as follows: “The results of this cross-sectional study suggests that the awareness of HPV among women visiting gynecological clinics in Kazakhstan is relatively high. However, the knowledge that HPV causes cervical cancer and awareness of HPV vaccination should be improved. As the success of the governmental programs for cervical cancer prevention largely depends on the population's knowledge and awareness of HPV and cervical cancer, there is an urgent need for educational intervention; both formal and informal. Population-based informational programs need to enhance the knowledge and awareness, which would increase understanding of the risks related to the HPV infection and its association with cervical cancer. Overall, this could improve the public's acceptance of the cervical cancer screening program in Kazakhstan.”

#2

Dear Reviewer, 

Thank you very much for the careful evaluation of our manuscript. We appreciate your time, efforts, valuable comments and suggestions that helped us to improve the manuscript quality. Please find below our point-by-point responses for all your comments.

Reviewer #2: Knowledge and awareness of HPV infection and HPV vaccine among Kazakhstani women

Issa et al present the findings from the well done study “Kazakhstani women’s Knowledge and awareness of HPV infection and HPV vaccine.”

The tables and figure are clear and concise.

The text, in contrast, especially in RESULTS is difficult to read and needs to be simplified. For example, is it necessary to repeat the data from the tables, especially that contained in parentheses (e.g., AOR=0.73.; CI: 0.39-1.34, referent=age group of 18-25).

L217: The text states that women age 46 and over were aware of HPV, which I take to mean this is an important finding. To me it then implies that younger women likely are not. Is it then necessary several lines later to actually state that younger women were not.

Response: Thank you very much for the comment. We are very sorry for the issues with the test. For non-native English speakers it is challenging. The manuscript text has been reviewed by the coauthor, Professor from Touro University, USA, Azliyati Azizan. The text was made more simple and clear. The results section was simplified. 

L235-L240 The text repeats data in results. The entire RESULTS section should be made more readable

Response: The results section was recomposed and repetitions in the text and tables were removed. 

L243: (Among those who were aware about HPV vaccine (N=1,215), less than half of the respondents (46.36%) knew that HPV is the major cause of cervical cancer).

If I interpret the authors’ meaning correctly, fewer than half of the 1215 women who were aware of the vaccine understood what the vaccine treated because they were not aware that HPV is the cause of cervical cancer. If so, the implication the implication for the need for education is important: What is cancer and how does it come about; what are vaccines and what do they treat.

Response: Thank you very much for the comment. We agree with the reviewer that the educational intervention is of a great importance and we highlighted it in the conclusion. 

L297-L304: Largely repeats the introduction. It should be truncated or eliminated. Much of the remaining DISCUSSION unnecessarily repeats the results and can be truncated.

Response: Thank you for the comment. The discussion part was revised to eliminate repetitions between the discussion and the introduction and results parts. Please see the manuscript submitted with the track changes.

The CONCLUSION section repeats too much of the results, but insufficiently says what was learned from the study.

Response: Thank you for the comment. The conclusion section was recomposed as was suggested and now appears as follows: “The results of this cross-sectional study suggests that the awareness of HPV among women visiting gynecological clinics in Kazakhstan is relatively high. However, the knowledge that HPV causes cervical cancer and awareness of HPV vaccination should be improved. As the success of the governmental programs for cervical cancer prevention largely depends on the population's knowledge and awareness of HPV and cervical cancer, there is an urgent need for educational intervention; both formal and informal. Population-based informational programs need to enhance the knowledge and awareness, which would increase understanding of the risks related to the HPV infection and its association with cervical cancer. Overall, this could improve the public's acceptance of the cervical cancer screening program in Kazakhstan.”

---

## [Editor Report · Decision Letter 1]

29 Nov 2021

Knowledge and awareness of human papillomavirus infection and human papillomavirus vaccine among Kazakhstani women attending gynecological clinics

PONE-D-21-11756R1

Dear Dr. Aimagambetova,

We’re pleased to inform you that your manuscript has been judged scientifically suitable for publication and will be formally accepted for publication once it meets all outstanding technical requirements.

Kind regards,

Stanley J. Robboy, MD

Academic Editor

PLOS ONE
---

## [Editor Report · Acceptance letter]

3 Dec 2021

PONE-D-21-11756R1 

Knowledge and awareness of human papillomavirus infection and human papillomavirus vaccine among Kazakhstani women attending gynecological clinics 

Dear Dr. Aimagambetova:

I'm pleased to inform you that your manuscript has been deemed suitable for publication in PLOS ONE. Congratulations! Your manuscript is now with our production department. 

Kind regards, 

on behalf of

Dr. Stanley J. Robboy 

Academic Editor

PLOS ONE